

# Evaluation of cross-generational exposure to microplastics and co-occurring contaminants on embryonic and larval behavior in fathead minnows, *Pimephales promelas*

Mackenzie Persinger[1,2] and Jessica Ward[1]

[1] Department of Biology, Ball State University, Muncie, IN, United States of America
[2] Current affiliation: College of Veterinary Medicine, Lincoln Memorial University, Harrogate, TN, United States of America

## ABSTRACT

Microplastics (MPs) are ubiquitous in freshwater systems, and understanding their effects on aquatic organisms is increasingly important. Microplastics also have unique surface properties that allow them to act as vectors for common environmental pollutants, such as endocrine-disrupting chemicals (EDCs), and can serve as an additional route of exposure to those contaminants. However, few studies have considered the cross-generational effects of microplastics on early life-stage behaviors in aquatic vertebrates. In this study, adult fathead minnows, *Pimephales promelas,* were exposed to microplastics alone ($MP_{Virgin}$) or in association with one of two environmentally relevant concentrations of 17 $\alpha$-ethinyl estradiol (low; $MP_{EE2\ 10}$ and high; $MP_{EE2\ 50}$) for 30 days and then were allowed to spawn. Embryonic activity was assessed for F1 offspring of MP-exposed adults at 4 days post-fertilization. After hatching, half of these larvae received continued exposure to MPs for 21 days, and larvae were tested in open-field swimming trials for swimming performance and space use at 14 and 21 days post-hatch. Before hatching, $MP_{EE2\ 10}$ F1 individuals showed reduced activity compared to other groups. After hatching, larvae from MP-exposed parents were more active than control larvae, but no changes in space use were observed. Evidence was limited for the effects of continued MP exposure on larvae after hatching, or combined effects of MPs and EE2. These results indicate that parental exposure to MPs induces subtle, sublethal effects on embryos and larvae that have the potential to affect individual fitness.

# INTRODUCTION

Anthropogenic environmental disturbances have negative impacts on wildlife in both terrestrial and aquatic environments. In aquatic habitats, examples of human-induced disturbances include changes in hydrology, habitat degradation, and pollutants in aquatic ecosystems (*Richmond, 1993*; *Rehage & Trexler, 2006*; *Magrini, Freitas & Uehara-Prado, 2011*; *Nearing et al., 2017*; *Jiang et al., 2020*). These anthropogenic changes may directly or

Corresponding author
Jessica Ward, ward@bsu.edu

indirectly alter ecosystem functionality by inducing stress at the individual level, ultimately impacting physiological processes, genetic diversity, and biodiversity (*Battisti, Poeta & Fanelli, 2016*; *Danehy & Moldenke, 2016*; *Almeida-Rocha et al., 2020*). One way that human activity negatively impacts ecosystems is by inducing changes in organismal behavior, including learning, foraging, social interactions, antipredator responses, and movement (*Candolin & Rahman, 2023*). Subtle, individual-level changes in behavior are increasingly recognized as drivers of ecosystem dynamics in human-modified environments because they can be magnified at the population and community levels (*Jacquin et al., 2020*; *Rahman & Candolin, 2022*). For example, habitat loss and fragmentation due to human interference alters juvenile dispersal in amphibians, which is a key factor maintaining population connectivity and persistence (*Cushman, 2006*). Moreover, individuals are often simultaneously exposed to multiple human-mediated environmental stressors, the behavioral outcomes of which may be complex and difficult to predict (*Schinegger et al., 2016*; *Lopez et al., 2023*).

However, growing evidence also suggests that not only do single and multiple anthropogenic stressors impact the current, directly affected generation, but they can have longer-lasting, trans-, cross-, and/or multigenerational effects on subsequent generations through epigenetic mechanisms, changes in the expression of proteins and hormone levels, and other processes (*Wolf & Wade, 2009*; *Wadgymar, Mactavish & Anderson, 2018*). For example, *Luu, Ikert & Craig (2021)* examined the combined effects of venlafaxine (a common aquatic pollutant) and increased water temperature and decreased dissolved oxygen (both of which are climate change stressors) on transcriptional and epigenetic responses of zebrafish in a multigenerational experiment. They found that exposure in the F0 generation to these combined stressors resulted in modifications in transcription of mRNA and microRNA in both F1 and F2 fish. Similarly, co-exposure to mercury and warmer water temperatures in the marine copepod (*Tigriopus japonicus*) for three successive generations led to mercury bioaccumulation that was significantly higher in co-exposed copepods and increased with successive generations, which consequently led to downregulation of several important processes such as protein metabolism and energy production (*Bai et al., 2023*). Changes in gene expression, metabolism, and other responses in organisms exposed to anthropogenic disturbances across multiple generations can lead to increased resistance or tolerance to these disturbances but ultimately decrease fitness, genetic diversity and resistance to secondary stressors as less-tolerant individuals are removed from the population with each generation (*Guttman, 1994*).

In recent decades, the presence of microplastics (MPs; *i.e.,* plastics smaller than <5 mm in diameter; *Chamas et al., 2020*) in the environment has become an area of research interest due to the potential impacts of exposure on organismal fitness (*Burns & Boxall, 2018*). Microplastics are introduced into freshwater environments from a variety of sources, including effluent, sewage runoff, and wastewater (*Lebreton et al., 2017*; *Wong et al., 2020*) and induce changes in the behavior of affected organisms in multiple fitness contexts, including predator–prey interactions (*De Sá, Luís & Guilhermino, 2015*; *Mannering, 2021*), reproduction (*Carter & Ward, 2024*), and foraging (*Uy & Johnson, 2022*). Exposure to MPs may impair individual performance through direct effects on body systems that regulate

behavior (for example, contaminant-induced changes in endocrine or neural signaling; *Hasan et al., 2024*). They may also serve as a vector for other organic pollutants that are easily adsorbed to the surface (*Wu et al., 2019*) and alter the behavior of individuals (*Luís et al., 2015*; *Bour et al., 2020*), such as pesticides, heavy metals, or endocrine-disrupting chemicals (*Zala & Penn, 2004*). In addition to direct effects on individual species, MPs may also indirectly influence the behavior of other species in the community, for instance by shifting competitive dynamics or modifying predation pressure (*Saaristo et al., 2018*; *Pan et al., 2022*).

Although substantial recent advances have been made in our knowledge of the complex behavior-mediated effects of MPs on organisms, several gaps remain to be filled. Notably, few studies have investigated how MPs—either alone or in combination with other environmental stressors—affect the behavior of aquatic organisms during critical early life stages (*e.g.*, embryos and larvae), despite the potential for behavioral impairments to reduce fitness and increase mortality (*Duan et al., 2020*). A second, related gap concerns the potential for cross-generational and/or multigenerational effects of exposure to MPs and co-occurring contaminants. Several studies have indicated transfer of MPs from mother to offspring in a variety of different aquatic organisms (*Pitt et al., 2018*; *Xia et al., 2023*; *Xue et al., 2023*). Parental exposure to MPs in combination with environmental contaminants can lead to the bioaccumulation of both MPs and contaminants in offspring and magnification of effects (*Li et al., 2022*). Although the mechanisms of MP cross-generational transfer are not fully clear (*Zhou et al., 2020*; *Yi et al., 2024*), research in fish has shown that MPs may be passed to offspring *via* the gametes and impair development, growth, and behavior—including through accumulation in the yolk sac of embryos (*Pitt et al., 2018*). Emerging research also suggests that MP exposure may induce changes in DNA methylation that are passed to subsequent generations *via* transgenerational epigenetic inheritance (*López de las Hazas, Boughanem & Dávalos, 2022*; *Wade et al., 2025*).

In this study, we exposed adult (F0) and larval (F1) fathead minnows, *Pimephales promelas* (a common freshwater fish used in ecotoxoicology studies), to MPs alone, or MPs previously immersed in 17 $\alpha$-ethinylestradiol (EE2), to evaluate the cross-generational behavioral consequences of MPs alone and combined with a EDC pollutant. 17 $\alpha$-ethinylestradiol is a synthetic hormone used in oral contraceptives, hormone-replacement therapies, and in the treatment of some cancers (*Adeel et al., 2017*). It is a common EDC in urban-impacted freshwater and marine environments that affects gonadal development and gene expression in both male and female fish that in turn affects reproduction, alters population sex ratios, and induces behavioral changes in individuals (*Vanden Belt, Verheyen & Witters, 2003*; *Aris, Shamsuddin & Praveena, 2014*). We chose EE2 for this study because its effects on the physiology, growth, reproduction, and behavior of fish at all life stages are well known (*Colman et al., 2009*; *Fenske et al., 2020*). Exposure to EE2 at the embryonic stage results in altered circadian rhythms and activity patterns (*Zhao, Zhang & Fent, 2018*). During the larval stage, exposure to EE2 alters behaviors such as swimming and neural gene expression (*Nasri et al., 2021*), and activity levels (*Bell, 2004*; *Soloperto et al., 2023*). Ultimately, the goals of this study were to (i) investigate whether there is an observable difference in the behavior of fish exposed to MPs alone *versus* those associated with EE2;

and (ii) test for potential cross-generational effects of MPs on the behavior of embryos and larvae in aquatic systems, alone or combined with EE2.

## MATERIALS & METHODS

### Experimental overview

Figure 1 shows an overview of the experimental design. To assess the potential for cross-generational effects of MPs, we first exposed male and female *P. promelas* in the adult (F0) generation to a clean water control (no MPs or EE2), pure MPs ($MP_{Virgin}$), or MPs in combination with one of two environmentally relevant concentrations of EE2 ($MP_{EE2\ 10}$: 10 ng/L or $MP_{EE2\ 50}$: 50 ng/L). The concentrations of EE2 used in the study were selected because they are environmentally relevant (*Aris, Shamsuddin & Praveena, 2014*; *Klaic & Jirsa, 2022*) and allow for comparison with previous studies (*Swank et al., 2022*; *Carter & Ward, 2024*). We designated F0 adults in the clean water control treatment as $F0^-$, and adults in the MP treatments as $F0^+$; the negative indicating no exposure to MPs or EE2, and the positive indicating that there was exposure to MPs and/or EE2. Fish used for breeding during the experiment were allowed to spawn before exposure to ensure that gamete production occurred during the exposure period. After spawning, the fish were housed separately by sex and underwent one of the four treatments for 30 days. After this period, males and females were paired for up to 14 days and eggs were collected. We recorded several metrics of reproductive success, including the numbers of pairs that produced eggs, the latency to spawning, and clutch size.

The effects of parental exposure were assessed on embryonic behavior at day 4 post-fertilization (dpf) using established metrics for embryonic toxicity testing (*von Hellfeld et al., 2023*). Immediately after testing, each clutch was split into two approximately equal-sized groups. One half of the clutch did not receive any further exposure to MPs and was labeled as ($F1^-$). Given the rapid embryogenesis and larval development of fathead minnows, it is likely that in nature offspring experience water quality conditions similar to those encountered by the breeding population. Therefore, the other half of each clutch continued with the same treatment as the F0 generation and was labeled ($F1^+$). For each of the three exposure treatments ($MP_{Virgin}$, $MP_{EE2\ 10}$, $MP_{EE2\ 50}$), two experimental groups of larvae were created: $F0^+/F1^-$, wherein only the F0 generation was exposed to MPs but not the larvae; $F0^+/F1^+$, wherein both the F0 and F1 generations were exposed to MPs. Control larvae were designated as $F0^-/F1^-$, wherein neither the F0 nor the F1 fish were exposed to MPs. For $F1^+$ larvae, dietary exposure to MPs began within 24 h of hatching and continued for 21 days; these larvae were fed a mixture of *Artemia* nauplii and MPs. $F1^-$ larvae were fed only *Artemia* nauplii. Subsets of larvae were dissected at 14 and 21 dpf to ensure consumption of MPs. Larvae were tested for both swimming performance and space use in open-field trials at 14 and 21 dph (days post-hatch). Swimming performance is a critical predictor of foraging success and predator evasion (*Fuiman et al., 2006*; *Nowicki, Miller & Munday, 2012*), and space use, particularly the propensity for individuals to enter the center of an open area (*Carlson & Langkilde, 2013*), is a well-established proxy for risky behavior and boldness (*Brown & Braithwaite, 2004*). We thus selected these behavioral endpoints
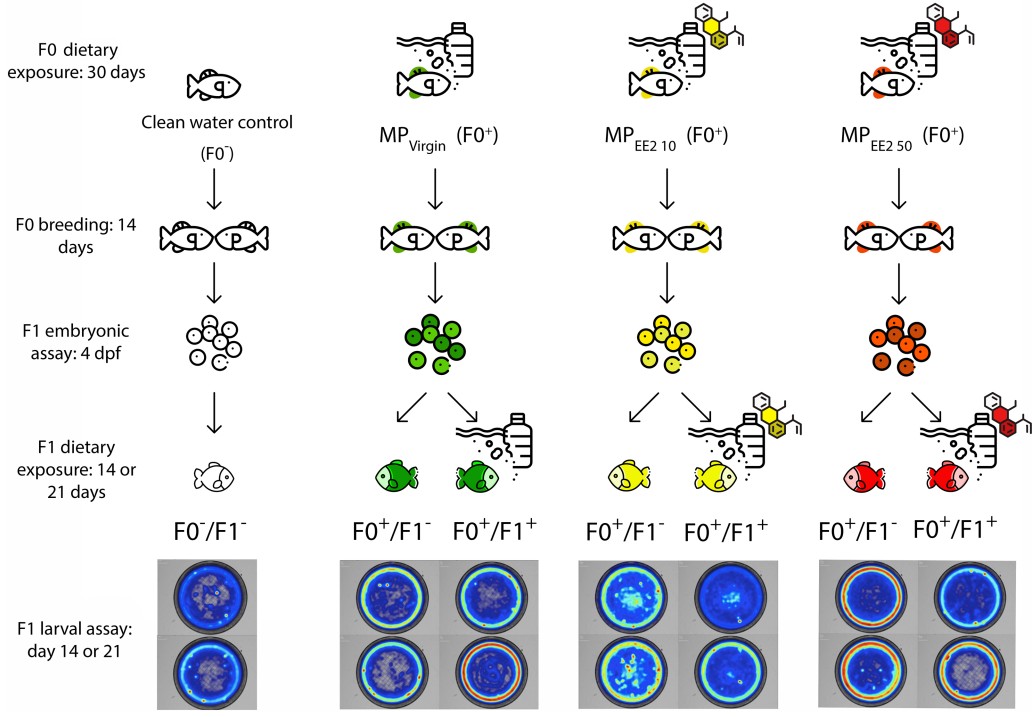

**Figure 1** **Diagram of the experimental protocol followed in this study.** Mature fathead minnows (F0) were exposed to one of four treatments (control; F0⁻ or MP_Virgin, MP_EE2 10, or MP_EE2 50; F0⁺) for 30 days, and then paired and permitted to spawn. The behavior of the resulting embryos was assessed on day 4 post-fertilization (dpf). Subsequently, each clutch was divided, and one half was reared to 21 dpf in the absence of any further exposure to MPs (F1⁻), and the other half was reared with continued exposure to the same treatment as the F0 generation (F1⁺). Larvae from each exposure configuration were tested in open-field behavioral assays at 14 and 21 dpf. Fish exposed to a clean water control are represented by no colors, fish exposed to the MP_Virgin treatment at any point are represented by the color green, MP_EE2 10 is represented by the color yellow, and MP_EE2 50 is represented by the color red. Heatmaps are representative examples from larval swimming performance and space use trials (day 14, top; day 21, bottom). Figure designed using resources from Flaticon.com.

because of their relevance to organismal fitness under natural conditions. All treatment and welfare procedures were performed in compliance with Ball State University's Institutional Animal Care and Use Committee (Approval # 1142896-17).

## General animal housing and care

A total of 570 fish were used for this study (130 adult breeding minnows (65 males and 65 females), and 440 larval minnows from 34 produced clutches). Prior to the start of the experiment, 6-month-old, reproductively mature fathead minnows (*Pimephales promelas*) were obtained from an aquatic culturing facility (Aquatic Biosystems, Fort Collins, CO, USA). We used *P. promelas* for this study because they are ubiquitous in North American freshwater systems (*Page & Burr, 2011*), have well-defined reproductive and developmental cycles (*United States Environmental Protection Agency, 1987*), and are extensively used in behavioral ecotoxicology studies (*Ankley & Johnson, 2004*), including those involving estrogenic EDCs (*e.g., Panter, Thompson & Sumpter, 1998*). The number of adult minnows

used reflected the number of pairings required to obtain a minimum of six genetically unique clutches per treatment (range: 6–11 clutches per treatment), to minimize the likelihood that variation among treatments was biased by parental characteristics. Upon arrival, adult fish were separated by sex and housed in a 530-L living stream (Frigid Units Inc, Living Stream System, Toledo, OH, USA) until use. Fish were kept under a 16:8-h light: dark cycle and at a water temperature of 23 °C to mimic summer breeding conditions. Water quality and the health of adult fish and subsequent offspring were monitored daily. Criteria established for euthanization of adult or larvae before the end of the experiment included evidence of significant injury, or illness as evidenced by symptoms such as lethargy, swimming or floating sideways, non-responsiveness, or laying on the bottom of the tank. However, no fish exhibited signs of injury or illness during the experiment. Fish were fed *Artemia* nauplii *ad libitum* twice a day, supplemented with blood worms once a day.

## Microplastics, chemical solutions, and treatments

In this study, we used virgin, spherical, transparent polyethylene microplastics (Cospheric, Santa Barbara, CA, USA). We chose polyethylene for this study because it is abundant in surface waters and is frequently consumed by aquatic organisms (*Azizi, Khoshnamvand & Nasseri, 2021*). The particles ranged in size from 150–180 µm in diameter, which is both environmentally relevant (*Conkle, Báez Del Valle & Turner, 2018*) and smaller than the maximum prey size consumed by *P. promelas* larvae (*United States Environmental Protection Agency, 1987*).

Fish were exposed to one of three dietary MP treatments in this study (*i.e.*, pure MPs ($MP_{Virgin}$), MPs associated with 10 ng/L EE2 ($MP_{EE2\ 10}$), or 50 ng/L EE2 ($MP_{EE2\ 50}$)), or to a clean-water control (no MPs or EE2). An EE2-only group was not included for two reasons; first, our primary goal was to determine whether there was evidence of behavioral differences in fish exposed to MPs with and without co-occurring contaminants (*Nobre et al., 2024*). Second, it was suggested that differences in EE2 exposure pathways (*e.g.*, *Al-Ansari et al., 2013*) could affect equivalency of exposure of EE2-only *vs* EE2-MP exposure groups. Estrogen-associated MPs for the $MP_{EE2\ 10}$ and $MP_{EE2\ 50}$ treatments were generated before the start of the experiment following previously described methods (*Swank et al., 2022*; *Carter & Ward, 2024*). In brief, appropriate amounts of 1-ug/mL EE2 stock solution were aliquoted into 1-L beakers of aged water. The MPs were placed into 75-micron mesh bags and immersed for 72 h. Solutions were freshly made and replaced every 24 h to account for any chemical degradation. After 72 h, the microplastics were removed from solution and allowed to air dry, after which they were aliquoted into falcon tubes and frozen at −20 °C until use. A similar protocol was followed for the $MP_{Virgin}$ treatment, except that MPs were immersed in a solution of cleaned, aged water only.

## Adult dietary exposure protocol and breeding

Adult minnows were introduced into 6-L tanks in an aquatic animal housing system (Pentair Aquatic Habitats, Apopka, FL, USA). The fish were housed in groups of 3–4, separated by sex. One exposure treatment was conducted at a time to prevent cross-contamination among tanks and because of space restrictions. Treatments were completed

in the order control, $MP_{Virgin}$, $MP_{EE2\ 50}$, $MP_{EE2\ 10}$ to eliminate the potential for inadvertent residual EE2 exposure by control and $MP_{Virgin}$ fish. Three exposure replicates per sex (*i.e.,* six exposure tanks in total) were conducted for the control group, three exposure replicates per sex were conducted for the $MP_{Virgin}$ treatment (six total tanks), seven exposure replicates were conducted for $MP_{EE2\ 10}$ treatment (14 total tanks), and five exposure replicates were conducted for $MP_{EE2\ 50}$ treatment (10 total tanks). Fish undergoing exposure treatment received MPs twice per day *via* an established dietary exposure protocol (*Swank et al., 2022*; *Carter & Ward, 2024*), with exposures separated by at least 7 h to ensure motivation to forage. At the start of each exposure event, the flow of water to the housing unit was turned off, and each tank was provided with two mL of live *Artemia* nauplii mixed with microplastics at a concentration of 100 MP/L. We selected this level of MP exposure based on reported estimates of MP concentrations in surface water samples from natural waterbodies (*Burns & Boxall, 2018*). The fish were permitted to forage naturally for 30 min. An air stone in each tank ensured that the microplastics remained in suspension (*Carter & Ward, 2024*). At the end of 30 min, the air stones were removed, and excess microplastics were removed using a microinvertebrate dip net. The flow of water to the housing unit was restored. Water turnover ensured that any residual remaining microplastics were removed within ~10 min, which were collected in a 75-micron mesh sock and discarded.

After the 30-day exposure period, one male and fish female fish were paired in each tank and permitted to breed for a maximum of 14 days. Pairs were discarded from the experiment if they did not produce a clutch during this timeframe, and the failure to breed was recorded. Ten breeding pairs were used for each of the control and $MP_{Virgin}$ treatments, all of which produced eggs. Twenty-eight pairs were used for the $MP_{EE2\ 10}$ treatment (17 of which did not spawn) and 17 pairs were used for the $MP_{EE2\ 50}$ treatment (9 of which did not spawn). The tanks were checked for eggs twice a day, and clutches were immediately removed when found and placed into 800-mL glass jars containing an air stone and 600 mL of aged, aerated water. Parental fish were only used once, to produce one clutch. After breeding, the fish were euthanized *via* an overdose of MS-222, measured for standard length, and preserved in 95% ethanol. The latency to spawning (*i.e.,* days since pairing) and total number of eggs laid in the clutch were also recorded.

## Embryonic development and behavior

Clutches were maintained in clean water at room temperature (25 °C) until hatching and checked twice a day for mortality. All clutches collected from F0 breeding pairs in the control, $MP_{Virgin}$, and $MP_{EE2\ 50}$ groups survived to hatching. In the $MP_{EE2\ 10}$ group, five clutches died prior to hatching. Thus, the final number of experimental clutches used was 10, 10, 6, and 8 for the control, $MP_{Virgin}$, $MP_{EE2\ 10}$ and $MP_{EE2\ 50}$ groups, respectively. For each clutch we recorded the minimum latency to hatching competence as the number of days from spawning to the time that the first egg hatched. Because researchers were aware of the exposure status of the test subjects, embryonic behavior was assessed at 4 dpf using automated behavioral tracking software (DanioScope, Noldus) to prevent the potential for researcher bias. To begin a trial, 10 embryos from a clutch were placed into a clear, acrylic dish. For a few clutches, only 4 or 5 embryos were used due to a small

number of individuals in the clutch. A thin layer of methyl cellulose on the bottom of the dish prevented the embryos from rolling out of view during the trial. Embryos were placed under a stereomicroscope (Stemi 508, Zeiss) and filmed with a microscope camera (Axiocam 208, Zeiss, Oberkochen, Germany) for 5 min. Embryonic activity during the trial was assessed as the percentage of time (activity %) the spent engaged in spontaneous activity (*e.g.*, flexing and rolling within the chorion), and the total number of distinct locomotor bursts performed.

## Larval dietary exposure protocol

After embryonic testing and just prior to hatching, each clutch was divided in half and the eggs were permitted to hatch. Beginning on the day of hatching and continuing for 21 days, one half of the F1 larvae from each clutch received continued F0 exposure ($F0^+/F1^+$ group) and the other half were maintained in clean water ($F0^+/F1^-$ group). Larvae were maintained in 800-mL jars equipped with an airstone under the same ambient environmental conditions as during the embryonic phase, and larval densities of $F0^+/F1^+$ and $F0^+/F1^-$ fish in jars were approximately equal for each clutch. Fish in the $F0^+/F1^+$ group underwent exposures twice daily for 30 min. At the start of an exposure, the larvae were provided with 100 uL of live *Artemia* nauplii mixed with microplastics of the appropriate treatment ($MP_{Virgin}$, $MP_{EE2\ 10}$, $MP_{EE2\ 50}$; 100 MP/L concentration). The $F0^-/F1^-$ group received 100 uL of *Artemia* nauplii only. After 30 min, the excess microplastics were removed using a microinvertebrate dip net and the bottom of each jar was vacuumed to remove debris. In addition, a 30% to 50% water change was performed daily to ensure that water quality remained high. To confirm that larvae ingested microplastics during natural foraging, we dissected a subset of randomly selected larvae ($n = 5$–29 individuals per treatment) on days 14 and on 21 under a stereomicroscope 30 to 60 min after feeding and counted the number of MPs observed in the reproductive tract.

## Larval open-field trials

Larval open-field swimming trials were conducted at 14 and 21 dph. Trials were conducted in a circular, transparent, Plexiglas area (3.5-cm diameter) with 2-mm x 2-mm gridlines on the bottom. The arena walls were tinted black using a marker to reduce outside visual disturbances. The arena was placed on a dimmable LED light board to provide illumination and centered under a GigE camera (Basler, AG, Germany) positioned 13 cm above the surface of the water. Each trial was conducted in a darkened room to minimize visual disturbance to the larvae. At the beginning of each trial, a single larva was placed into the arena containing aged, aerated water and allowed to acclimate for 1–2 min. The activity of the larva was then recorded for 5 min, and swimming behavior was analyzed using automated behavioral tracking software (Ethovision XT, Noldus, Wageningen, the Netherlands, ver. 13.0), which prevents observer bias. No more than five larvae from each clutch were used for trials (range: 1–5), and each larva was only used once. Individuals were immediately euthanized after trials *via* an overdose of MS-222 and preserved in 95% ethanol. Trials were conducted in a quiet, darkened room at 25 °C to reduce the potential for disturbance.

The arena was divided into two zones for video imaging analysis; a 1.6-cm diameter central zone in the middle of the arena with a zone-exit threshold of 2 mm, and a surrounding outer zone that included the edges of the arena. Spatial measurements were calibrated for each video before analysis using a 2-mm arena grid. Metrics of swimming performance that we measured included the total distance moved during the trial (mm), mean swimming velocity (mm/s), and the duration of trial time spent actively swimming (s). Space use was measured and compared among treatments as the duration of trial time spent in the central zone (s) and the number of visits made to the central zone. At the end of the experiment, all remaining individuals not used in trials were euthanized *via* an overdose of MS-222.

## Statistical analyses

Non-parametric Kruskal–Wallis tests were used to compare differences among groups in the duration of embryonic development, clutch size, and time to spawn. Spearman rank correlations were used to evaluate the relationship between female body length and clutch size, and among behavioral variables. Preliminary analyses indicated that our measures of embryonic behavior were significantly and positively correlated (Spearman $r = 0.95$, $P < 0.001$), as were larval behavioral variables for swimming performance (Spearman $r = 0.98$–$0.99$, all $P < 0.001$) and behavioral space use (Spearman $r = 0.88$–$0.92$, all $P < 0.001$) on testing days 14 and 21. We therefore used principal component analysis (PCA) to combine embryonic behavioral variables into one behavioral index score for use in statistical analysis, and similarly used PCA to combine variables for larval swimming performance and space use, respectively, on each of days 14 and 21. Principal component analysis yielded one significant component for each model. The components explained more than 96–98% of the variation in embryonic behavior and larval swimming performance, respectively, and 73–76% of the variation in space use.

Generalized estimating equation models (GEE) were conducted to assess differences among treatments in embryo activity, with PCA score used as the dependent variable and parental treatment set as the fixed factor. Generalized estimating equations were also used to analyze larval swimming performance and space use across treatments. For these analyses, PCA scores were used as the dependent variable and parental treatment and larval exposure (continued exposure after hatching or not) were included as fixed factors, as well as the interaction between these two factors. Data collected on days 14 and 21 were analyzed separately, and all models were fit with a robust estimator for the covariance matrix and an independent working correlation structure. Clutch identification was specified as the subject variable, and individual was specified as the within-subject variable to account for correlations among individuals originating from the same clutch. In cases where significant factor effects were found, least significant difference (LSD) pairwise *post-hoc* tests were conducted to evaluate the differences among factor levels. No *a priori* criteria were set for excluding data from analysis and all data were included, with the necessary exception that adult breeding pairs that didn't produce a clutch were excluded from F1 analysis. Analyses were conducted in R (ver 4.3.1) or SPSS (ver 29, IBM Corp., Armonk, NY, USA).

## RESULTS

### Preliminary dissections

Preliminary dissections confirmed that larvae did ingest microplastics *via* the dietary exposure protocol. Microplastics were observed in the gastrointestinal tract of randomly selected larvae on both days 14 and 21. On day 14, 20% (1/5) of larvae in the $F0^+/F1^+$ $MP_{Virgin}$ group contained MPs; 41% (11/27) of larvae in the $F0^+/F1^+$ $MP_{EE2\ 10}$ group contained MPs; and 40% (6/15) of larvae in the $F0^+/F1^+$ $MP_{EE2\ 50}$ group contained MPs. Of those with MPs, mean ($\pm$ SD) numbers of particles were as follows: $F0^+/F1^+$ $MP_{Virgin}$: $1 \pm 0$, $F0^+/F1^+$ $MP_{EE2\ 10}$: $2.09 \pm 1.51$, and $F0^+/F1^+$ $MP_{EE2\ 50}$: $1.67 \pm 1.63$. On day 21, 56% (15/27) of $F0^+/F1^+$ $MP_{Virgin}$ larvae had MPs in their digestive tract; 41% (12/29) of $F0^+/F1^+$ $MP_{EE2\ 10}$ larvae contained MPs; and 54% (14/26) of $F0^+/F1^+$ $MP_{EE2\ 50}$ larvae contained MPs at the time of dissection. The average numbers of MPs in the $F0^+/F1^+$ $MP_{Virgin}$, $MP_{EE2\ 10}$, and $MP_{EE2\ 50}$ groups on day 21 were $3.8 \pm 5.61$, $5.08 \pm 9.80$, and $6.36 \pm 7.94$ MPs, respectively. We previously confirmed that reproductively mature fish readily ingest MPs during dietary exposures (*Swank et al., 2022*), so the F0 breeding fish were not dissected for this study.

### F0 reproduction and early life history

We did not find any significant effects of exposure on the latency to spawning in F0 breeding pairs ($X^2 = 5.44$, *df* = 3, *P* = 0.14), clutch size ($X^2 = 7.27$, *df* = 3, *P* = 0.06), or the duration of development before first evidence of hatching competence ($X^2 = 7.71$, *df* = 3, *P* = 0.05) (Fig. 2). However, both the spawning latency and the developmental time to hatching competency showed a U-shaped trend (Figs. 2A, 2C). On average, the time to spawning for reproductively mature F0 fish in the control, $MP_{Virgin}$, $MP_{EE2\ 10}$, and $MP_{EE2\ 50}$ treatments was $7.6 \pm 3.7$ days, $5 \pm 4.2$ days, $4.3 \pm 4.3$ days, and $9.9 \pm 6.9$ days. The duration of development to the earliest evidence of hatching competence in the control, $MP_{Virgin}$, $MP_{EE2\ 10}$, and $MP_{EE2\ 50}$ treatment groups was $6 \pm 1.2$ days, $5 \pm 1.2$ days, $4.7 \pm 0.8$ days, and $6 \pm 1.1$ days, respectively. No significant correlations were found between female standard length and clutch size for any treatment (F0 control: $r = -0.26$, *P* = 0.47; $MP_{Virgin}$: $r = -0.02$, *P* = 0.96; $MP_{EE2\ 10}$: $r = -0.26$, *P* = 0.61; $MP_{EE2\ 50}$: $r = -0.54$, *P* = 0.16). However, clutch size showed a linear, negative, dose-dependent trend (Fig. 2B), with clutches in the control group being on average approximately 26% larger than those in the $MP_{Virgin}$ group, and 31% and 58% larger than clutches in the $MP_{EE2\ 10}$ and $MP_{EE2\ 50}$ groups, respectively.

### Embryonic activity

Sample sizes for embryos tested from the four treatments are as follows: $n = 94, 83, 55, 70$ embryos from the $F0^-$ control group (from 10 clutches), $F0^+$ $MP_{Virgin}$ (eight clutches), $F0^+$ $MP_{EE2\ 10}$ (six clutches), and $F0^+$ $MP_{EE2\ 50}$ (eight clutches). We found a significant overall main effect of parental treatment on embryonic activity ($X^2 = 18.88$, *df* = 3, *P* < 0.001). *post-hoc* tests indicated that embryos whose parents were exposed to $MP_{EE2\ 10}$ had reduced locomotor activity compared to offspring from parents in the control, $MP_{Virgin}$ and $MP_{EE2\ 50}$ groups (all *P* < 0.007; Fig. 3).

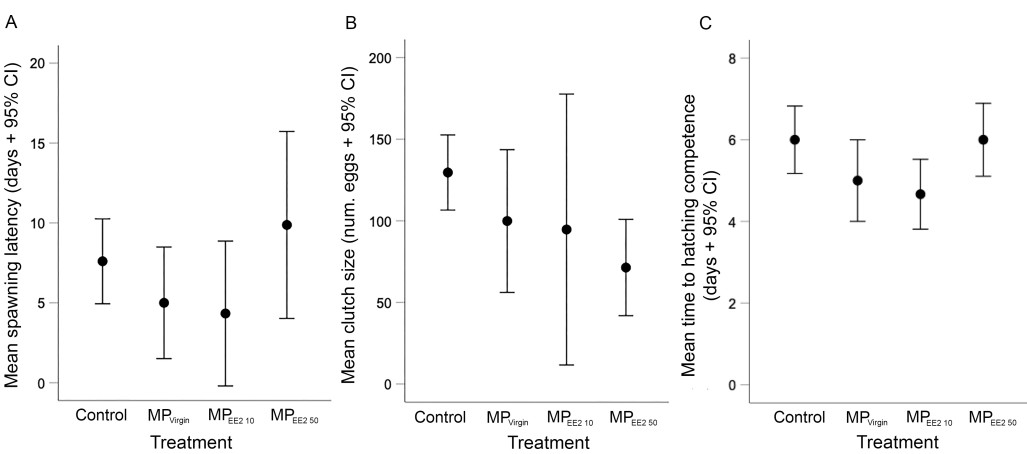

**Figure 2** **Effects of F0 exposure to microplastics on spawning activity and egg development.** There was no statistically significant effect of exposure on any spawning or developmental variable. (A) Latency to spawning among F0 adults exposed to microplastics alone ($MP_{Virgin}$) for 30 days, or to MPs soaked in a low ($MP_{EE2\ 10}$) or high ($MP_{EE2\ 50}$) concentration of 17-$\alpha$ ethinyl estradiol (EE2). Points and whiskers represent the mean number of days $\pm$95% CIs from conclusion of the exposure period (day of pairing) to the production of a clutch. (B) Mean clutch size produced by F0 breeding pairs after 30 days of exposure. Points and whiskers represent the mean number of eggs $\pm$95% CIs in clutches produced from breeding pairs in each treatment. (C) Duration of time until first evidence of hatching competence of F1 eggs spawned by F0 adults in different treatments. Points and whiskers represent mean number of days from spawning to evidence of first hatching and 95% CIs. Sample sizes for (A) (*i.e.,* breeding pairs), (B) and (C) (clutches) are as follows: Control $n = 10$; $MP_{Virgin}$ $n = 8$; $MP_{EE2\ 10}$ $n = 6$; $MP_{EE2\ 50}$ $n = 8$.

## Larval swimming performance

Parental (F0$^+$) exposure to MPs significantly influenced swimming behavior of F1 offspring on both day 14 ($X^2 = 11.46$, $df = 3$, $P = 0.009$) and day 21 ($X^2 = 19.44$, $df = 3$, $P < 0.001$). Pairwise *post-hoc* tests revealed that on both days, F0$^+$ larvae from MP-exposed parents swam further, faster, and spent more time engaged in swimming activity than larvae from control parents (F0$^-$) (Figs. 4A, 4B). The magnitude of difference between control and exposed fish was generally more pronounced at day 21, with offspring from MP-exposed parents swimming approximately 35% to 43% farther and 35% to 42% faster compared to non-exposed embryos during performance trials (F0$^+$/F1$^-$ and F0$^+$/F1$^+$ groups averaged for each MP treatment). We did not detect a significant effect of continued post-hatch larval exposure (F1$^+$) on swimming performance at either day 14 ($X^2 = 0.73$, $df = 1$, $P = 0.39$) or day 21 ($X^2 = 1.44$, $df = 1$, $P = 0.23$), or evidence of an interaction between these two factors (day 14: $X^2 = 1.33$, $df = 2$, $P = 0.52$; day 21: $X^2 = 0.06$, $df = 2$, $P = 0.97$), suggesting few additive effects of indirect exposure during gametogenesis and direct exposure after hatching.

## Larval space use

We did not find a significant effect of parental (F0$^+$) exposure on how individuals used the arena space on either day 14 ($X^2 = 6.89$, $df = 3$, $P = 0.08$) or day 21 ($X^2 = 5.71$, $df = 3$, $P = 0.13$), nor a significant interaction between parental treatment and larval exposure on either day (day 14: $X^2 = 3.57$, $df = 2$, $P = 0.15$; day 21: $X^2 = 3.45$, $df = 2$, $P = 0.18$).

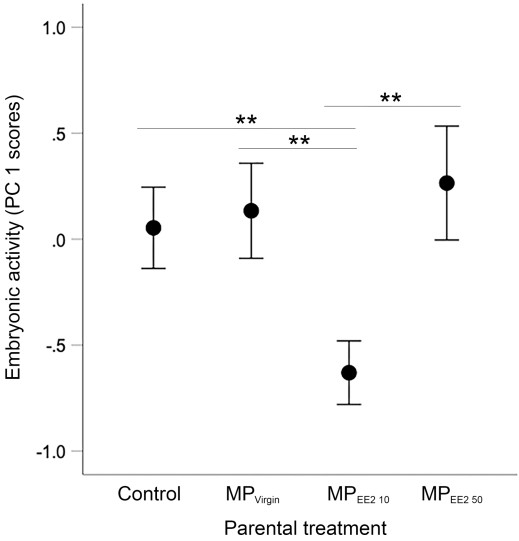

**Figure 3 Effects of F0 exposure on F1 embryonic activity at 4 dpf.** Generalized estimating equations revealed a significant effect of parental (F0) treatment on spontaneous embryonic activity. Points and whiskers represent mean principal component analysis (PCA) index scores ±95% CIs derived from embryonic behavioral variables (% of trial time spent active and number of discrete locomotor bursts). Significant pairwise differences among marginal means from the main effect of parental (F0) treatment are shown with asterisks; ** < 0.01. Embryo sample sizes in each treatment are as follows: Control $n = 94$; $MP_{Virgin}$ $n = 83$; $MP_{EE2\ 10}$ $n = 55$; $MP_{EE2\ 50}$ $n = 70$.

On day 14, larvae from MP-exposed parents who received continued exposure visited and spent more time in the center of the center of the arena compared to those exposed only *via* parental gametes ($X^2 = 7.16$, $df = 1$, $P = 0.007$), but this effect disappeared by day 21 ($X^2 = 2.71$, $df = 1$, $P = 0.10$) (Figs. 4C, 4D).

## DISCUSSION

For aquatic organisms, there is still limited information regarding the effects of microplastics on fitness and behavior at critical early life stages, their potential to act as vectors for common aquatic environmental pollutants, and the effects of exposure on subsequent generations. In this study, we found that parental (F0) exposure to MPs alone and/or in combination with EE2 had detectable effects on the behavior of F1 individuals at both the embryo and larval stages, and comparatively limited evidence of further behavioral changes associated with continued exposure after hatching or combined effects of MP and EE2 exposure compared to MP exposure alone. These data provide new insights about the potential for MPs to exert cross-generational effects on organismal fitness and behavior during early life stages.

Although the differences between the groups did not reach statistical significance, some trends were observed in early life history metrics that are consistent with studies of other aquatic contaminants. Larvae in the $MP_{Virgin}$ and $MP_{EE2\ 10}$ groups showed first evidence of hatching that was earlier than that of embryos reared in the control treatment, similar to previous work that found that exposure to some contaminants reduces the duration of

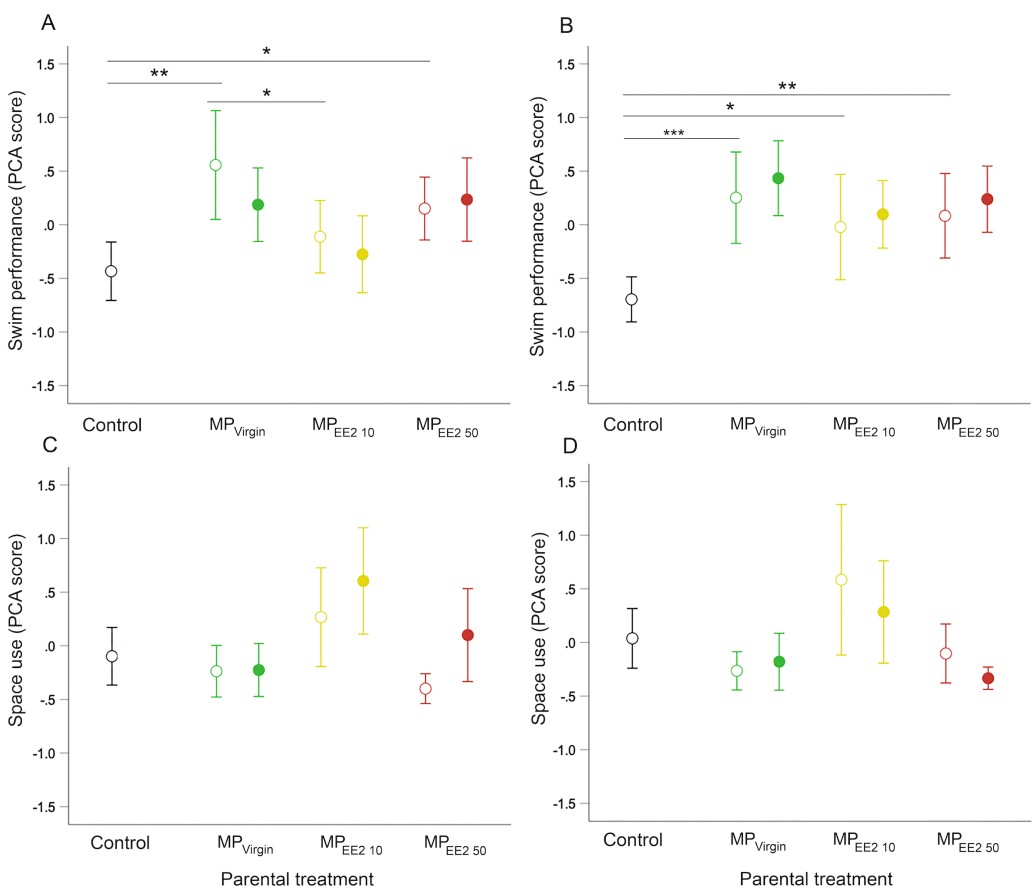

**Figure 4  Effects of F0 and F1 exposure on swimming performance and space use on days 14 (left column) and 21 (right column) post-hatch.** Points and whiskers represent mean principal component analysis (PCA) index scores and 95% CIs. Exposure status during early larval development is indicated by shape: open circle = parental exposure only, and filled circle = parental and larval exposure. (A–B) Swimming performance on (A) day 14 and (B) day 21. (C–D) Space use on (C) day 14 and (D) day 21. Generalized estimating equation models revealed a significant overall main effect of parental treatment (control, $MP_{Virgin}$, $MP_{EE2\ 10}$, and $MP_{EE2\ 50}$) on swimming performance in open-field trials; lines and asterisks represent significant pairwise differences among marginal means from the main effect of parental (F0) treatment; * < 0.05, ** < 0.01, *** < 0.001. Sample sizes are as follows; day 14: control $n = 45$, $MP_{Virgin}$ $F0^+/F1^-$ $n = 24$, $MP_{Virgin}$ $F0^+/F1^+$ $n = 29$, $MP_{EE2\ 10}$ $F0^+/F1^-$ $n = 30$, $MP_{EE2\ 10}$ $F0^+/F1^+$ $n = 30$, $MP_{EE2\ 50}$ $F0^+/F1^-$ $n = 32$, $MP_{EE2\ 50}$ $F0^+/F1^+$ $n = 32$; day 21: control $n = 45$, $MP_{Virgin}$ $F0^+/F1^-$ $n = 24$, $MP_{Virgin}$ $F0+/F1^+$ $n = 29$, $MP_{EE2\ 10}$ $F0^+/F1^-$ $n = 26$, $MP_{EE2\ 10}$ $F0+/F1^+$ $n = 30$, $MP_{EE2\ 50}$ $F0^+ +/F1^-$ $n = 32$, $MP_{EE2\ 50}$ $F0+/F1^+$ $n = 32$.

embryogenesis and induces premature hatching (*e.g.*, EE2 in *Fundulus heteroclitus*, *Peters et al., 2010*). Premature hatching has also been observed in marine medaka larvae whose parents were exposed to polystyrene microplastics (*Wang et al., 2021*). We also observed a linear dose-dependent-like decrease in mean clutch size across treatments, with F0 fish co-exposed to MPs and the highest concentration of EE2 having the smallest mean clutch size compared to nonexposed adults. These data suggest that exposure to MPs, both alone and in combination with EE2, have the potential to affect the number of offspring produced, and that MPs may potentially act as a vector for locally co-occurring contaminants known

to affect fecundity. Exposure to EE2 at concentrations ranging from 10–500 ng/L have been shown to decrease spawning and fecundity in various fish species (*e.g.*, *Vanden Belt, Verheyen & Witters, 2003*; *Lee, Lin & Chen, 2014*). In fish, parental quality can also affect survival, clutch size, and other offspring life-history traits (*Kerrigan, 1997*; *Green & McCormick, 2005*). For example, reproductively mature *Acanthochromis polyacanthus* fed a low-quality diet had smaller clutches and lower offspring survival rates than those fed a high-quality diet (*Donelson, McCormick & Munday, 2008*). Because the number and quality of offspring produced directly affect recruitment of the next generation (*Beldade et al., 2012*; *Saenez-Agudelo et al., 2015*), these findings have implications for the fitness and sustainability of aquatic species exposed to aquatic pollutants.

We observed differences in the behavior of both embryos and larvae from some MP-exposed parental groups compared with those from non-exposed parents (*Crain, Kroeker & Halpern, 2008*; *Serra et al., 2020*). Before hatching, embryonic activity showed a non-linear pattern of response across treatments, with those in the $MP_{EE2\ 10}$ group showing reduced activity compared to individuals in the other treatments. Such non-linear U-shaped dose–response patterns are commonly reported in ecotoxicological studies involving estrogenic EDCs (*Calabrese & Baldwin, 2001*; *Pamplona-Silva, Mazzeo & Bianchi, 2018*), and may reflect several factors, including opposing effects on the body at different doses mediated by multiple receptors with different affinities, negative feedback systems at higher doses, and dose-dependent changes in metabolism (*Lagarde et al., 2015*). However, after hatching larvae from MP-exposed parents were consistently and significantly more active than control larvae on both days 14 and 21. Such changes in swimming performance have potential to directly affect predation mortality (*Domenici & Blake, 1997*; *Anwar et al., 2016*; *Krause et al., 2017*) by increasing the conspicuousness of individuals from prey species, or altering the kinematics and effectiveness of predator evasion maneuvers (*Mesa et al., 1994*). For example, changes in velocity that alter the expression of the C-Start maneuver—a highly conserved, innate evasive response (*Gillette, 1987*; *Domenici & Blake, 1991*)—have been shown to increase predation-related mortality (*Walker et al., 2005*). Taken together our data suggest that parental exposure to MPs (alone or in combination with EE2) affects locomotor activity in F1offspring both before and after hatching although the consequences for fish fitness remain to be tested.

We did not find a significant effect of parental MP exposure on space use on either testing day, but we did find limited evidence that direct exposure to MPs altered boldness soon after hatching, which is relevant to foraging, as well as interactions with predators and conspecifics (*Biro & Stamps, 2008*; *Budaev & Brown, 2011*). Generally, more bold individuals outperform less-bold conspecifics in foraging performance but also may experience higher rates of predation (*Ward et al., 2004*; *Ólafsdóttir & Magellan, 2016*). Relatively few studies have evaluated the effects of MPs on boldness, but those that have generally found that MP-exposure increased risk-taking behaviors in fish (*McCormick et al., 2020*; *Gorule et al., 2024*). Interestingly however, other studies have shown reduced or altered boldness in Siamese fighting fish (*Dzieweczynski et al., 2014*; *Hebert et al., 2014*) and three-spined stickleback (*Gasterosteus aculeatus*: *Dzieweczynski & Greaney, 2017*) exposed to EE2. Microplastics can serve as routes of exposure for other aquatic pollutants, potentially

magnifying or otherwise modifying the effects of either MPs and/or the contaminants on organismal physiology and behavior (*Wen et al., 2018*; *Qu et al., 2019*). Thus, it is possible that the effects of EE2 and MPs on individuals are potentially agonistic with respect to boldness. This possibility highlights the difficulty in predicting the overall fitness consequences of human-induced behavioral changes (*Coleman & Wilson, 1998*; *Ólafsdóttir & Magellan, 2016*).

A main finding of this study is that direct exposure to MPs in one generation can have observable cross-generational effects. Few studies have evaluated the cross-generational effects of MP and EE2 co-exposure (*Henriksson, 2017*), but evidence of F0 exposure has been shown in F1 offspring for both contaminants alone. For example, *Volkova (2015)* showed that zebrafish and guppies exposed to EE2 at early life stages demonstrated increased anxiety-like behaviors, which persisted through two consecutive subsequent generations. Parental exposure to MPs has also been shown to alter growth, physiology, hatching, and swimming performance in offspring (*Wang et al., 2019*; *Bringer et al., 2022*). However, cross-generational effects of MPs co-exposed with other environmental pollutants are well documented (*Fang et al., 2016*; *Nogueira et al., 2022*; *Junaid et al., 2023*). For example, exposure of F0 *Daphnia magna* to MPs and glyphosate (a common pesticide) induced negative effects in both F1 and F2 offspring (*Nogueira et al., 2022*). Cross-generational or multigeneration exposure has been implicated in modified gene expression, delayed development, and increases bioaccumulation in offspring (*Junaid et al., 2023*), with implications for recruitment, reproduction, and survival (*Fang et al., 2016*; *Cormier et al., 2022*).

The responses of larvae in this study who were exposed to MPs *via* the parents only were mostly similar to those of larvae exposed indirectly *via* parents *and* directly after hatching, suggesting that further direct exposure of the offspring after hatching did not increase the effect of pre-hatching exposure. One explanation for this finding could the comparative immaturity of the organ systems during embryogenesis compared to after hatching, including those involved in detoxification (*Azad, 2013*). Such an explanation would be consistent with general evidence that aquatic organisms are generally more sensitive to contaminants at earlier life stages compare to later ones—and especially so when critical developmental processes are underway (*Black et al., 1982*; *Hutchinson, Solbe & Kloepper-Sams, 1998*; *Foekema et al., 2012*). Although the mechanisms by which the effects of exposure in the F0 generation were passed to subsequent generations are currently unknown, prior research suggests that MPs may be directly transferred to offspring *via* the gametes, eliciting transcriptional changes and triggering developmental toxicity in offspring (*Pitt et al., 2018*; *Cormier et al., 2022*). Alternatively, parental exposure may impair gamete quality and induce epigenetic changes that influence gene expression and phenotype in subsequent generations (*Yaripour et al., 2021*; *Wade et al., 2025*; reviewed in *Yi et al., 2024*). From an evolutionary perspective, such changes may represent either adaptive or maladaptive responses that shape evolutionary trajectories of subsequent generations (*Donelan et al., 2020*; *Castano-Sanz, Gomez-Mestre & Garcia-Gonzalez, 2022*). Thus, more deliberate assessments of how parental effects contribute to organismal responses in

human-altered environments are key to predicting both ecological and evolutionary outcomes for affected species.

## CONCLUSIONS

Taken together, the results of this study suggest that cross-generational exposure to MPs induces subtle, sublethal effects on early-life stage behaviors, with the potential to affect individual fitness. Moreover, the effects of MP exposure on larvae are comparable with or without co-exposure to EE2. Although care must be taken in extrapolating the results of controlled, lab-based studies such as this to more complex natural environments, these results underscore the importance of considering parental effects as a key mechanism by which environmental stressors influence phenotypic variation across generations. Further research is now needed to (i) fully understand the effects of multigenerational exposure of MPs and their potential to act as vectors for common environmental pollutants, and (ii) identify the specific mechanisms involved. Such information is central to efforts to understand and predict the extent to which individual-level performance might impact population-level processes and community-level effects.

## ACKNOWLEDGEMENTS

We would like to thank Travis Beckett, Sam Horton, Jewel Johnson, Brooke Karasch, Dylan Mann, Karly Steinberg, Leah Sodo, Leah Turner, and Abby Yake for their assistance with animal care and experimental maintenance. We are also grateful to three anonymous reviewers for their thoughtful suggestions and comments on an earlier version of our manuscript.

### Funding

This work was funded by a Ball State ASPIRE Student Research Grant to Mackenzie Persinger, and an Illinois-Indiana Sea Grant to Jessica Ward. The funders had no role in study design, data collection and analysis, decision to publish, or preparation of the manuscript.

### Grant Disclosures

The following grant information was disclosed by the authors:
Ball State ASPIRE Student Research Grant.
Illinois-Indiana Sea Grant.

### Competing Interests

The authors declare there are no competing interests.

### Author Contributions

- Mackenzie Persinger conceived and designed the experiments, performed the experiments, analyzed the data, prepared figures and/or tables, authored or reviewed drafts of the article, and approved the final draft.

- Jessica Ward conceived and designed the experiments, analyzed the data, prepared figures and/or tables, authored or reviewed drafts of the article, and approved the final draft.

## Animal Ethics

The following information was supplied relating to ethical approvals (*i.e.*, approving body and any reference numbers):

The Ball State University Institutional Animal Care and Use Committee approved the study (1142896-17).

## Data Availability

Data is available at Dryad:

Ward, Jessica; Persinger, Mackenzie (2025). Data from: Evaluation of cross-generational exposure to microplastics and co-occurring contaminants on embryonic and larval behavior in fathead minnows, Pimephales promelas [Dataset]. Dryad. https://doi.org/10.5061/dryad.w6m905r00.

## Supplemental Information

Supplemental information for this article can be found online at http://dx.doi.org/10.7717/peerj.19927#supplemental-information.

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
