# Peer review of "Evaluation of cross-generational exposure to microplastics and co-occurring contaminants on embryonic and larval behavior in fathead minnows, Pimephales promelas"

_PeerJ, doi:10.7717/peerj.19927_

## Round 0.1 · original submission · Major Revisions

Three experts have reviewed your manuscript and identified a number of issues that require a fundamental revision of the manuscript. The most important aspects are (1) the lack of a group exposed only to EE2, which is necessary to demonstrate additive or even synergistic effects, and (2) a more detailed rationale for the chosen experimental design, including the choice of EE2 as test substance. Further points for revision are detailed in the reviews, addressing inter alia the statistical analysis, inconsistencies between the presentation and interpretation of results in the text and figures and also the lack of detail in the figure descriptions.

I hope that the reviewers' comments will enable you to fundamentally revise the manuscript, which is a prerequisite for its acceptance.

Reviewer 1 ·

Basic reporting

Though this manuscript benefits with considering cross generation, complex experimental design and heavy workload, it can not be considered for acceptance due to the significant drawbacks.
(1) There is no EE treatment group.
(2) Microplastics exist in the freshwater and also in the food. Why only very short period was used for the exposure? Such exposure way is not environmentally relevant.
(3) The experiment design is so complicated, but there is only one concentration of MPs with two different EE concentrations. Generally, it is still a great challenge to conduct toxicology experiments with microplastics due to their unique features (e.g., uneven distribution in the water). The complicated design and limited concentration groups make the results of effects less convincing. What is more, not all the endpoints showed the similar or dose-dependent changes.
(4) The presentation is not concise or clear enough. It is hard for the readers to grasp the key points due to so many different groups marked with various symbols.

Experimental design

no comment

Validity of the findings

no comment

Reviewer 2 ·

Basic reporting

I loved this study on the transgenerational effects of MP and EE2 pollutants on fish behavior. The chosen treatments are justified. The experimental design is relevant and very elegant, with adequate sample sizes. Statistical analyses are adequate to answer the research questions. Results are complex but clear, and show that MP and EE2 induce hyperactivity in F1, both in eggs and after hatching. I have some comments and suggestions below:

Summary L33: larvae received continuous exposure of what?

The introduction is interesting and well written, but I feel that the behavioural part is not enough developed. I suggest shortening some parts on ecological effects or generalities on pollutant effects, and developing how behavioural changes can have cascading effects on fitness, evolution and ecosystems (to strengthen the rationale for your study), develop how pollutants can have either direct (neurotoxic) or indirect effects on behavior (physiological/metabolism and trade off changes), and then go towards multistress effects that are poorly known (how some pollutants can have synergistic or antagonistic effects, like MP and EDC in your case). Then develop your multigenerational approach which is really cool and promising to better understand long term effects (like you said).
I understand that you explained these points in your introduction, but they are not enough highlightened and sometimes a bit “lost” in some generalities on pollution.

L 110, you state that data suggest that MP have potential transgenerational effects. Please explain a bit more about this!! There are debates about MP, if they reach the blood stream and organs or not, if they affect behavior or not, and how, with various results depending on the study/model/multistress etc. Please present these studies/results to better highlight the originality and importance of your cool study. For instance L 106: how are MP transferred through generations?

Also, you don’t justify enough the choice of EE2 in the intro, please explain why you chose this particular pollutant and how you think that your choice of MPxEE2 is the best choice to better understand how pollution affects aquatic wildlife (there is ample info in the methods but I think some elements are needed as well in the intro)

L135 and hereafter: Is this representative of urban rivers? Or other anthropogenic contexts? (I suppose urban context close to wastewater effluents?).

Experimental design

Research question is well defined and methods appropriate to test the research questions with ample sample size and replicates.

Fig 1 is really neat and the experimental design is perfectly relevant (and elegant I may say). I would suggest to add sample size on Fig 1 and legend and method text. Also make the legend a bit more developed.

L142 explain what you measured in adult responses (spawning success etc)

L147. I like that you exposed offspring to the same stressors as parents. Please justify this in terms of environmental relevance (adaptation of offspring to parental environment) and justify this in light of the environmental realism and timing (pollution levels quite constant in the wild, so that it will reflect local (mal)adaptation)

L 152 and the whole paragraph. Maybe I missed it but please show some validation of this treatment efficiency based on previous studies? Or your measurement of MP and EE2 measurement within the fish tissue? To justify that your treatment is efficient. I see now L332 that you validated the treatment yourself after dissection, just explain this briefly in the methods.

Fig 2: write the result inside the legend as well (no significant differences, differences etc) L306 statistical analyses seem relevant and adequate (lmer)

Validity of the findings

Results and Discussion

Results are convincing and I loved the discussion. Clear and interesting, with relevant comparison with the existing literature.

I could not control the original dataset but statistical analyses are justified.

Conclusions are linked to results.

L 516-528. Here I would like to understand more clearly what mechanism of transgerenational effects are likely in your study. And which further studies would be necessary to test it.

It is surprising (and cool) that parental effects are stronger than the exposure of offspring themselves. Please explain a bit more how you see this and offer some interpretations.
I think that you also should interpret some of your results in a more evolutionary framework: parental effects/adaptation/maladaptation across generations and expected effects for the evolutionary trajectories of aquatic wildlife.

Reviewer 3 ·

Basic reporting

The article is written in clear and unambiguous professional English throughout although some minor mistakes were detected. Specifically, at line 174, a word is missing which makes it hard to understand the full meaning of the phrase: six genetically … clutches per treatment …

The figures used are high quality and relevant but some figures lack details in the figure description. In Figures 4 and 5 it is unclear which groups are being labelled as statistically significant with brackets and asterisks. It looks as if the F0+/F1- and the F0+/F1+ treatments are being considered as a single entity for statistical analysis but this is not mentioned neither in the figure description nor in the text and needs to be clarified. If this is the case, it would also be important to mention why the two groups are being pooled together. Furthermore, in figure 4A, B, C, treatment D14, the Control and MP EE2 10 groups look as they could be significantly different but are not indicated as such. Please check your statistics and make sure to not have forgotten a significance bracket and asterisks.

Experimental design

No comment

Validity of the findings

I have two comments regarding the discussion section:
- Embryonic activity, locomotor bursts, distance moved, average velocity and time spent moving at day 14 all show opposing trends between the MP EE2 10 and MP EE2 50 groups. This is very evident in the figures but is never addressed in the discussion. In line 415 you say that microplastics exposure caused trends towards increased activity both in embryos and larvae but this is not represented by the MP EE2 10 group which shows the exact opposite trend. This needs to be mentioned and contextualised.

- In lines 416- 419 it is mentioned how the effects shown by the MP EE2 50 group might suggest a synergistic effect of MP with EE2. I don’t think such a statement can be made without having assessed the individual effects of each compound first, and since there was no EE2 only group in your study I don’t think you should include such a statement. From what we know, the effects seen in the MP EE2 50 group could be just the sum of the individual effects of MP and EE2. The same statement is repeated in line 456.

---

## Round 0.2 · Minor Revisions

We are almost there! Reviewer 2 suggests a few more changes, including to clarify some ambiguities. I agree that this would improve the clarity of the manuscript and its figures. I am happy to accept the manuscript for publication after the appropriate updates.

Reviewer 2 ·

Basic reporting

The authors took into account all my comments and I found this version much imrproved. I still have some comments and thus recommend minor revisions.

L 109: MPs may be passed to offspring via the gametes? In the yolk sac of embryos you mean? Thus only in females?

L 114. Maybe specify here that your main research question is not underpinning mechanisms but rather transgenerational behavioural consequences of MPs combined with EDC pollutant.

I would clarify again a bit more. Goal 1: test whether MP can act as a vector for EDC (such as EE2). Goal 2: test for transgenerational effects of MPs combined with EE2.
And you can definitely not test antagonistic or synergic multistress effects of MPs and EE2, given that you don’t have controls with EE2 alone (which is not a problem to me since it was not an hypothesis that you wanted to test, but in this case be very careful and clear about that)

L 130. I don’t see the difference between objective i) and ii). Maybe pool them.

Results. Generally results are quite complex and not easy to follow, although authors did a good job at analyzing them and discussing them.

I suggest some editing of the figures could be somehow improved further for clarity. For instance Fig 4 and 5 could be clarified by using the same colors as in the experimental design Fig 1 for instance. Please also put all * indicating significant differences always above bars (and not sometimes above sometimes below for day 14 vs 21).

I would also recommend to use PCA to analyze some correlated behaviours such as distance moved, time spent moving, average velocity and angular velocity since they are likely correlated (sorry I didn t suggested this in my previous review). Idem for frequency of entries to central arena, total time in central arena, and latency to first entry to central arena). I realize it’s quite tedious to rerun the statistics but such multivariate analyses such as PCA and extract synthetic variables could be complementary or better than your current analyses variable by variable and bring a clearer synthetic view of your behavioural changes across treatments and generations. At the very least I would like to know whether and how all the measured behavioural traits are correlated with each other and if yes why you did not use multivariate analyses, especially since you have many complex results.

L 433. Do F1 groups affected by treatments correspond to F0 parents with decreased clutich size of fecundity? More generally, is there a link between your results on F0 and on F1?

L 458 L 461 I don’t understand what biphasic means please rephrase.

L 478. Yes and I would remain cautious by ending up with a sentence like “although the consequences for fish fitness remains to be tested”. More generally some sentences and conclusions could be toned down a little.

L 480. Hence my question about potential correlations between behaviours (do more active fish are also those that are less exploratory and /or less bold?).

L 483 493. I would tone down this part on personality, since you don’t measure repeated and stable behaviours aka personalities. You can compare to literature on personality but stay cautious about your results since measures are not repeated across contexts and may not reflect personality differences.

L 495-499. Again, too speculative and not cautious enough, remove or rephrase more cautiously

L 500 505. To me this is the main point of your study (MP acting as a vector of EE2), rather than the trade offs between traits and fitness consequences of your behavioural changes observed.

L 514. And this cross generational effects as well.

L 529 It is even more interesting since F0+/F1+ groups are similar as F0+/F1- groups, showing that exposure in F0 only is sufficient to induce F1 changes whithout esposing the F1 themselves. This is really cool.

L532. Not sure I agree (or maybe I misunderstood so please rephrase). F1 may be more sensitive than F0 but it does not explain that F0+/F1- group respond as strongly as the F0+/F1+ group. It would just explain why F1 would respond more strongly than adult F0 on their behaviour.

L550 not sure its reflects perception of risk. Rephrase.

Conclusion needs to be improved. You discuss here some mechanisms that should be discussed earlier in the transgenerational part of the discussion. The conclusion could rather focus on a synthetic view of your results and their implications for current knowledge, especially on MP as vectors and potential transgenerational effects of MP+EDC multipollution on wildlife.

Experimental design

no comment

Validity of the findings

see above

Additional comments

see above

Reviewer 3 ·

Basic reporting

No comment

Experimental design

No comment

Validity of the findings

No comment

Additional comments

I think that the article has been significantly improved. All the points I raised on the previous manuscript have been addressed in the text. The text and figures are much clearer and easier to understand. The results and discussion section have also significantly improved and they argument well for the results of this study. I have nothing more to add.

---

## Round 0.3 · accepted · Accept

Thank you for the thorough revision of the manuscript. I hereby certify that you have adequately taken into account the reviewers' comments and improved the manuscript accordingly. Based on my assessment as an Academic Editor, your manuscript is now ready for publication.